# Improving sexually transmitted infection screening, testing, and treatment among people with HIV: A mixed method needs assessment to inform a multi-site, multi-level intervention and evaluation plan

**Kathleen Cullinen**[⬤]*[◐], **Macsu Hill**[◐], **Taylor Anderson**[¤◐], **Veronica Jones**[‡], **John Nelson**[⬤][‡], **Mirna Halawani**[‡], **Peijia Zha**[⬤][‡]

François-Xavier Bagnoud Center, School of Nursing, Rutgers, The State University of New Jersey, Newark, New Jersey, United States of America

◐ These authors contributed equally to this work.
¤ Current address: The University of Queensland Ochsner Clinical School, Brisbane, Queensland, Australia
‡ VJ, JN, MH and PZ also contributed equally to this work.
* kmc366@sn.rutgers.edu

## Abstract

Bacterial sexually transmitted infections (STIs) continue to be a worsening public health concern in the United States (US). Though the national incidence of HIV infection has decreased over recent years, that of chlamydia, gonorrhea, and syphilis have not. Despite national recommendations on prevention, screening, and treatment of these STIs, these practices have not been standardized. Nine Health Resources and Services Administration Ryan White HIV/AIDS Program funded clinics across 3 US jurisdictions (Florida, Louisiana, and Washington, DC), were selected as clinical demonstration sites to be evaluated in this mixed method needs assessment to inform a multi-site, multi-level intervention to evaluate evidence-based interventions to improve STI screening and testing of bacterial STIs among people with or at risk for HIV. These 3 US jurisdictions were selected due to having higher than national average incidence rates of HIV and bacterial STIs. Descriptive statistics and deductive analysis were used to assess quantitative and qualitative needs assessment data. Results indicate the following needs across participating sites: inconsistent and irregular comprehensive sexual behavior history taking within and among sites, limited routine bacterial STI testing (once/year and if symptomatic) not in accordance with CDC recommendations, limited extragenital site gonorrhea/chlamydia testing, limited annual training on STI-related topics including LGBTQ health and adolescent/young adult sexual health, and limited efforts for making high-STI incidence individuals feel welcome in the clinic (primarily LGBTQ individuals and adolescents/young adults). These findings were used to identify interventions to be used to increase routine screenings and testing for bacterial STIs.

**Data Availability Statement:** All relevant data are within the paper and its Supporting Information files.

**Funding:** John A. Nelson, PhD, CNS, CPNP (JAN) - Principal Investigator This study was funded by the U.S. Department of Health and Human Service (HHS) Health Resources and Services Administration's (HRSA) HIV/AIDS Bureau (HAB) and Bureau of Primary Health Care (BPHC) under grant number U90HA32147. HAB: https://hab.hrsa. gov/ BPHC: https://bphc.hrsa.gov/ Through a cooperative agreement, the funders had a role in study design and the decision to publish the manuscript.

**Competing interests:** The authors have declared that no competing interests exist.

# Introduction

Bacterial STIs continue to be a worsening public health concern in the United States (US). Though the national incidence of HIV infection has decreased over recent years, this is not the case for common bacterial STIs (ie, chlamydia (CT), gonorrhea (GC), and syphilis) [1, 2]. Bacterial STIs have associated morbidities and mortalities [1] (eg, infertility, chronic inflammation and pain, congenital anomalies, neonatal death, and neurocognitive disease), increase the risk of HIV transmission from a non-virally suppressed person [3], and contribute to an ever-increasing public health burden. In the case of GC, curbing the growth of antimicrobial resistance is also a public health priority [4]. Despite national recommendations on screening and treatment of STIs in people with HIV or at risk for HIV, disparities exist in the regular screening, treatment, and prevention of STIs among this population [5]. It is critical to identify STIs early, to treat as per the Centers for Disease Control and Prevention (CDC) established guidelines, and to follow-up regarding the effectiveness of the treatment and possible reinfection. Addressing the many barriers across institutions, communities, providers, and patients associated with providing STI care in HIV care clinics is an integral step towards reducing STI incidence.

At the institutional level, limitations include insurance reimbursement for laboratory testing of extragenital site specimens, integration of routine bacterial STI screening practices into clinical demonstration sites (CDS) not already providing this care, and limited availability for walk-in appointments at these facilities. Community barriers include perceived non-confidentiality and social stigma associated with STIs and sexual behaviors. It is known that stigma associated with health conditions, especially those relating to sexual health, make it difficult for patients to access care [6]. Providers must help patients address concerns of perceived discomfort or reluctance with sexual history taking and genital exams [5], increase their knowledge of screening and treatment recommendations, reduce time constraints related to comprehensive STI screening, and minimize patient test cost burden. Training providers on stigma, as well as making clinics more welcoming or inviting to patients, especially those identifying as lesbian, gay, bisexual, transgender, questioning or queer (LGBTQ), may also reduce resistance to seeking routine STI care [5, 7, 8]. Patient-level barriers may include potential breach of confidentiality, inaccessibility of clinical services, and absence of provider recommendation for STI screening [2, 5, 9]. Despite the existence of "Ask, Screen, Intervene," an HIV/STI transmission prevention intervention for providers to use with people with HIV [10], and other such programs, gaps exist in the clinical integration of CDC recommendations for STI testing and treatment for populations disproportionately affected by STIs in and outside of HIV care [11, 12].

These gaps are thought to contribute to recent national and statewide incidence trends. Based on the CDC's 2018 Sexually Transmitted Diseases Surveillance Report [1], Washington, DC, had the highest rates of CT, GC, and primary and secondary syphilis among the 50 states. Louisiana ranked second for the highest rate of CT, fifth for the highest rate of GC, and seventh for the highest rate of primary and secondary syphilis. Florida ranked eighth for the highest rate of primary and secondary syphilis, and 29th for both CT and GC incidence [1]. Louisiana and Florida ranked third and seventh, respectively, for congenital syphilis. In 2016, Washington, DC, had the highest HIV incidence and prevalence in the country, followed by Florida and Louisiana [13].

Considerable efforts have been made to identify opportunities to improve STI screening, testing, and treatment in populations disproportionately impacted by STIs. Unfortunately, the implementation of evidence-based interventions outside of STI-specific clinics has been limited [14]. The need for guidance on emerging evidence-based interventions may account for

the wide variation in clinic-specific operational procedures. To this end, a 3-year Health Resources and Services Administration (HRSA) Special Project of National Significance (SPNS) was launched to evaluate the implementation and efficacy of evidence-based interventions in 9 selected CDSs. A pre-intervention mixed method needs assessment was done in year 1 of this 3-year project. In this paper, we describe the methods and findings of the phase 1 needs assessment used to inform the design of our phase 2 multi-site, multi-level intervention and evaluation plan.

## Mixed methods

This study served as a baseline assessment to inform the multi-site, multi-level implementation evaluation for this HRSA-funded project to improve STI screening, testing, and treatment among people with or at risk for HIV. It consisted of a mixed method needs assessment that included a primary qualitative thematic analysis. Quantitative survey data was examined and triangulated with qualitative themes to improve interpretation. Five quantitative assessment tools and in-depth interviews were employed to measure the need for STI interventions within 9 CDSs across 3 US jurisdictions.

Five assessment tools were administered to key CDS staff deemed relevant to the project. Key CDS staff included a clinical prescriber (eg, MD, DO, NP, PA), a clinical non-prescriber (eg, RN, SW, MA), and a designated Change Champion (ie, a clinical prescriber or clinical non-prescriber who serves as the lead for the study at each CDS). The clinical prescriber and clinical non-prescriber were identified by the designated Change Champion. Triangulated findings of the 5 assessment tools and interviews determined which interventions, from a list of 9 evidence-based interventions identified by a federally convened Technical Experts Panel, would be implemented and how they would be evaluated throughout phase 2 of the study. The 9 evidence-based interventions, organized by the 3 categories of training, clinical, and non-clinical, included the following:

### Training

1. Provider Training on administering and discussing sexual health histories [2], and culturally competent care [15].

2. Training for Peer Navigators and non-clinical staff to increase STI screening [16].

### Clinical

3. Express testing to increase STI testing in clinical sites [17].

4. Patient self-collected STI testing for GC and CT that educates providers, decreases provider time, and increases patient's ease could increase testing [5].

5. Home/Alternative Location Testing for STIs to reduce the stigma of clinic visits, make testing locations more convenient, and minimize time constraints [18].

### Non-Clinical

6. Task Shifting or delegating tasks to less specialized staff (eg, moving responsibilities of a task from a prescriber, such as a DO, MD, NP, or PA, to a nurse, medical assistant, peer navigator, or other non-licensed staff) has been shown to be increase routine HIV and STI screening [19].

7. Computer Assisted Self-Interviews for STI risk assessment has been associated with identifying high risk behaviors more commonly, less time spent by the provider taking a sexual health history and has been highly acceptable when used by patients [20].

8. Strategic placement of specimen collection materials such as swabs near materials for other routine procedures, such as pap tests, has been demonstrated to be effective by making it easier for providers to collect samples [21].

9. Welcoming clinic spaces have been shown to be particularly important for two populations that have high rates of STIs, youth and LGBTQ populations [7].

This study was approved by the Institutional Review Board at Rutgers University. Consent forms with Rutgers University contact information were distributed to all subjects following explanation of the surveys and prior to their participation in the study. Upon completion of the surveys, subjects received contact information to request an aggregate summary of study findings.

## Setting

The study took place in 9 Ryan White HIV/AIDS Program (RWHAP) funded HIV care clinics across 3 US jurisdictions (Florida, Louisiana, and Washington, DC) with higher-than-average incidence rates of HIV and bacterial STIs. Three CDSs were selected within each of the 3 jurisdictions based on their willingness to participate in the study. Two of the 9 RWHAP CDSs are also HRSA Bureau of Primary Health Care (BPHC) funded Health Centers, providing care to patients with HIV as well as those at risk for HIV.

## Data source

### Qualitative assessment tool.

1. *Clinical Team Member Interview*: Three semi-structured interviews at each of the 9 CDSs were conducted with a clinical prescriber (eg, MD, DO, NP, PA), a clinical non-prescriber (eg, RN, SW, MA), and the designated Change Champion onsite and audiotaped, without personal identifiers, in a private room designated by the CDS. Interviews were completed by 2 investigators trained and published in qualitative research. Investigator 1, PhD, CNS, CPNP, is the Principal Investigator of this study and co-author of this paper. Investigator 2, PhD, MPH, CHES, is the Director of this study and a co-author of this paper. Investigators asked interviewees questions from a pre-written script that focused on current clinic GC/CT and syphilis policy and procedures; barriers to and supports for bacterial STI screening, testing, treatment, and follow-up among people with HIV and/or at risk for HIV; and clinical staff cultural competence. Audiotapes of responses were transcribed, and thematic analysis was completed using NVivo V12.0. The average duration of the interviews was 45 minutes.

### Quantitative assessment tools.

2. *Pre-Intervention Data Survey (2016–2017)*: Administered online via Qualtrics and disseminated to a designated Change Champion at each of the 9 CDSs to develop an understanding of each CDS's patient population and the number of patients tested, diagnosed, and/or treated for GC, CT and/or syphilis during 2016 and 2017. Additionally, the volume and quality of data provided by this survey was scrutinized by researchers to determine if new or additional data collection methods were necessary to capture robust

quantitative data relating to patient demographics, and GC, CT and syphilis screening, testing, and treatment at each CDS for the phase 2 evaluation study.

3. *Clinical Team Member Process, Attitudes, & Beliefs Survey*: Administered online via Research Electronic Data Capture (REDCap) to 3 individual clinical team members consisting of the designated Change Champion, a clinical prescriber, and a clinical non-prescriber at each of the 9 CDSs. This survey was designed to gather quantitative data from each CDS that relates to 9 evidence-based interventions being considered. Questions were chosen to address the 3 categories of potential interventions (training, clinical, and non-clinical). Survey response categories consisted of single and multiple-choice questions as well as Likert scale questions.

4. *STI Screening Readiness Checklist*: Administered onsite to a composite team of 3 clinical team members consisting of the designated Change Champion, a clinical prescriber, and a clinical non-prescriber. Survey response categories consisted of 1) Yes; 2) No; and 3) I don't know for each given statement designed to assess clinical team member feedback on clinical experiences related to STI prevention, testing, treatment, and follow-up.

5. *Clinic Workflow Operations Checklist*: Administered onsite to a composite team of 3 clinical team members consisting of the designated Change Champion, a clinical prescriber, and a clinical non-prescriber. This survey was designed to provide researchers with an understanding of each CDS's patient flow. The Change Champion, a clinical prescriber, and a non-clinical prescriber from each CDS were asked a series of questions of yes/no questions about whether key clinical operations relevant to bacterial STI care are performed at their CDS.

## Participants

From each of the 9 CDSs, a Change Champion, a clinical prescriber, and a clinical non-prescriber was designated by clinic administration to complete the *Pre-Intervention Data Survey (2016–2017)*, the *Clinical Team Member Process, Attitudes, & Beliefs Survey*, the *STI Screening Readiness Checklist*, the *Clinic Workflow Operations Checklist*, and the onsite *Clinical Team Member Interview* for a total sample size of 27 per assessment tool.

## Thematic analyses

Interview transcript coding was completed by 2 investigators trained in qualitative research. Codes from the *Clinical Team Member Interview* were inductively generated to fall into the 3 intervention categories, training, clinical, and non-clinical, to guide theme generation. The resultant themes included 1) training barriers and recommendations (ie, when gaps in cultural knowledge are evident or when it is mentioned that more training on cultural sensitivity is needed); 2) clinical barriers and recommendations (ie, when a subject describes clinical challenges that affect bacterial STI care and provides clinical recommendations to improve routine bacterial STI care); and 3) non-clinical barriers and recommendations (ie, when a subject describes non-clinical challenges that affect bacterial STI care and provides non-clinical recommendations to improve routine bacterial STI care).

Coding was done independently and with frequent communication between investigators to generate common code definitions. The kappa coefficient was calculated to be 0.96. Quantitative survey data was analyzed using descriptive statistics. All data was summarized as frequency and percentage. Resultant themes were triangulated with aggregate survey responses

that further supported thematic evidence. Survey results were analyzed using mixed methods utilizing SPSS V25.0 and NVivo V12.0 software programs.

## Results

The *Clinical Team Member Interview* and the 4 quantitative surveys were completed by each Change Champion, clinical prescriber, and clinical non-prescriber from each CDS. However, one interview was not recorded leaving a total of 26 interviews that were transcribed and included in the analysis.

All 9 change champions completed the *Pre-Intervention Data Survey (2016–2017)*. **Table 1** shows the results of the *Pre-Intervention Data Survey (2016–2017)* which provided an estimate of the baseline numbers for the subpopulations tested, diagnosed, and/or treated for GC, CT and/or syphilis at each of the CDSs during 2016 and 2017. These subpopulations will be evaluated in the phase 2 evaluation study: individuals at risk for HIV; young adults; transgender women; MSM; and pregnant individuals with HIV.

There was a 100% response rate from 27 Change Champions, clinical prescribers, and/or clinical non-prescribers who completed the *Clinical Team Member Process, Attitudes & Beliefs Survey* and represented the 9 CDSs. This survey consisted of 5 components to include 1) sexual history taking, 2) STI testing, 3) STI treatment, 4) clinical barriers to STI testing and treatment, and 5) non-clinical barriers to STI testing and treatment. Main findings are presented in **Table 2**. Although 59% of the CDSs offer patient self-collection for GC/CT NAAT specimens, 44% of respondents report patient refusal of provider GC/CT NAAT collection (oropharyngeal, rectal, and/or genital). In addition, 44% of CDSs conduct a consistent, comprehensive sexual history on intake; and 18% of the respondents reported provider discomfort with sexual history taking and the patient specimen collection process. Twenty six percent (26%) of the respondents reported their CDS' clinic space as being less than friendly to LGBTQ individuals and adolescents/young adults. Finally, 37% of the respondents reported the culture of their CDS as less than culturally competent for both LGBTQ individuals and adolescents/young adults. Compared to clinical non-prescribers, clinical prescribers demonstrated a larger need for cultural competence training.

Differences were noted in responses between clinical prescribers, clinical non-prescribers, and Change Champions. More clinical prescribers and Change Champions (67% each) than clinical non-prescribers (44%) reported that their CDSs offer patient self-collection for GC/CT NAAT specimens; more clinical prescribers (78%) than clinical non-prescribers (22%) and Change Champions (33%) reported patient refusal of provider GC/CT NAAT collection (oropharyngeal, rectal, and/or genital); more clinical non-prescribers (67%) than clinical prescribers and Change Champions (33% each) conduct a consistent, comprehensive sexual history on intake; both clinical prescribers and Change Champions (22% each) reported more provider discomfort with sexual history taking and the patient specimen collection process than clinical non-prescribers (11%); more clinical prescribers (33%) than clinical non-prescribers and

**Table 1. Pre-intervention data survey (2016–2017).**

| Jurisdiction | People with HIV | People at risk of HIV | MSM with HIV | Adolescents/ Young Adults | Pregnant Individuals with HIV | Transgender Women with HIV |
|---|---|---|---|---|---|---|
| Florida | 2600 | 0 | 757 | 128 | 58 | 31 |
| Louisiana | 2007 | 1500 | 277 | 287 | 71 | 6 |
| Washington, DC | 731 | 90 | 85 | 70 | 2 | 4 |
| **Overall Total** | **5338** | **1679** | **1119** | **485** | **131** | **41** |

**Table 2. Clinical team member process, attitudes & beliefs survey.**

| | Respondents % | Reported Findings |
|---|---|---|
| **Sexual History Taking** | 44 | Conduct a consistent, comprehensive sexual history on intake |
| | 74 | Conduct follow-up sexual histories at acute care visits when symptomatic for an STI |
| **STI Testing (among sexually active adolescents and adults with HIV)** | 67 | Test for STIs (syphilis and GC/CT at least one anatomical site) on at least an annual basis |
| | 18 | Test for STIs every 3–4 months (syphilis and GC/CT at least one anatomical site) |
| | 78 | Test for STIs if symptomatic for an STI |
| | 59 | Offer patient self-collection for GC/CT NAAT |
| **STI Treatment** | 52 | Bring back patients into clinic for a positive STI test result after being tested within 1–3 days |
| | 48 | Bring back patients into clinic for a positive STI test result after being tested within 4–10 days |
| **Clinical Barriers to STI Testing and Treatment** | 44 | Patient refuses to have provider do NAAT collection (oropharyngeal, rectal, and/or genital) |
| | 26 | Patient refuses to provide urine for NAAT |
| | 18 | Provider discomfort with sexual history taking and specimen collection process |
| | 15 | Supplies for STI testing are not easily accessible in exam rooms |
| **Non-Clinical Barriers to STI Testing and Treatment** | 26 | On a scale of 1 to 5 (1) very unfriendly, (2) unfriendly, (3) neutral, (4) friendly, and (5) very friendly), 25% of the CDSs rated their CDS as *less than friendly* (ie, scale of 1–3) to LGBTQ individuals |
| | 26 | On a scale of 1 to 5 (1) very unfriendly, (2) unfriendly, (3) neutral, (4) friendly, and (5) very friendly), 26% of the CDSs rated their CDS as *less than friendly* (ie, scale of 1–3) to adolescents/young adults |
| | 37 | On a scale of 1 to 5 (1) very culturally incompetent, (2) culturally incompetent, (3) neutral, (4) culturally competent, and (5) very culturally competent), 37% of the CDSs rated their CDS as *less than culturally competent* (ie, scale of 1–3) for both LGBTQ individuals and adolescents/young adults |

Change Champions (22% each) reported their CDS' clinic space as being less than friendly to LGBTQ individuals; more clinical non-prescribers (33%) than clinical prescribers and Change Champions (22% each) reported their CDS' clinic space as being less than friendly to adolescents/young adults; more clinical prescribers and Change Champions (44% each) than clinical non-prescribers (22%) reported the culture of their CDSs as less than culturally competent for LGBTQ individuals; and more clinical prescribers and clinical non-prescribers (44% each) than Change Champions (22%) reported the culture of their CDSs as less than culturally competent for adolescents/young adults. However, it must be noted that the designated Change Champion at each CDS may also serve as a clinical prescriber or clinical non-prescriber so response breakdown by role must be cautiously considered.

The *STI Screening Readiness Checklist* was administered onsite at each of the 9 CDSs to the clinical team including the Change Champion, a clinical prescriber, and a clinical non-prescriber. There was a 100% response rate from the 9 CDSs. Main findings are presented in **Table 3**.

The results of the *Clinic Workflow Operations Checklist* are presented in **Table 4**. The *Checklist* was designed to provide an understanding of each CDS's patient flow through the clinic. The Change Champion, a clinical prescriber, and a non-clinical prescriber at each CDS responded to a series of questions about whether key clinical operations relevant to bacterial STI care are performed at their CDS, in addition to non-clinical barriers to and supports for STI testing and treatment. All 9 CDSs reported that 100% of their providers conduct a sexual history, and 100% of patients are asked to provide urine for GC/CT NAAT. Among patients, 67% self-collect specimens for GC/CT NAAT. Among providers, 89% collect or request oropharyngeal and rectal specimens for GC/CT NAAT, 56% collect genital specimens for GC/CT NAAT, and 78% discuss HIV testing, if needed. Eighty-nine percent (89%) of CDSs conduct patient satisfaction assessments after each visit, quarterly, and/or annually per CDS policy, but none of the assessments ask specifically about satisfaction with STI-related care. The *Clinic*

**Table 3. STI screening readiness checklist.**

| Number of Clinics Reporting Yes for Each Indicator (%) | Indicator |
|---|---|
| 9 (100) | Staff knowledge of STI screening, testing, diagnosis, and treatment |
| | Clinic capacity to increase GC, CT, and syphilis testing |
| | Provider time to conduct physical exams for indicators of STIs |
| | Provider knowledge to conduct physical exams for indicators of STIs |
| | Having the supplies needed for GC, CT, and syphilis testing |
| | Working to reduce identified barriers related to STI testing, diagnosis, treatment, and follow-up |
| 8 (89) | Laboratory testing of extragenital site GC/CT NAAT specimens along with urine or genital site NAAT specimens |
| | Having a policy and procedure for providing necessary follow-up care and support to patients diagnosed with an STI |
| 7 (78) | Having a way to systematically monitor STI testing, diagnosis, treatment, and follow-up data for clinic population(s) |
| | Providing routine STI harm-reduction counseling (condom use, sex with drug use, U = U) to all patients |
| 6 (67) | Having the capacity to provide HIV and STI testing and treatment services to partners of people at risk of HIV |
| | Having the supplies needed for HIV testing[a] |
| 4 (44) | Implementing policies and procedures by clinic staff to allow for maximum reimbursement of STI services provided |
| | A process in use to evaluate patient care satisfaction and/or experiences regarding STI testing and treatment |
| 3 (33) | Having policies and procedures in place regarding staff member(s) responsibility for prevention of HIV (for HIV-uninfected patients), GC, CT, and syphilis |
| 9 (100) | State or local Department of Health (DOH) provision of Disease Intervention Specialist (DIS) services for syphilis |
| | State or local DOH provision of DIS services for HIV |
| 3 (33) | State or local DOH provision of DIS services for GC and CT |
| 8 (89) | Walk-in appointments for STI testing or treatment can be easily accommodated on the same day |
| 6 (67) | Utilizing a range of media platforms to communicate STI information to MSM |
| 5 (56) | Utilizing a range of media platforms to communicate STI information to pregnant individuals |
| 4 (44) | Utilizing a range of media platforms to communicate STI information to adolescents/young adults |
| | Utilizing a range of media platforms to communicate STI information to people at risk for HIV |
| 1 (11) | Utilizing a range of media platforms to communicate STI information to transgender women |

[a]100% of BPHC-funded Health Centers.

*Workflow Operations Checklist* also identified non-clinical barriers to and supports for bacterial STI testing and treatment. Among the CDSs, 44% of waiting rooms have visible indications of LGBTQ support such as a rainbow flag, a designated safe space sticker, images of same-sex couples on educational materials, and/or images of transgender affirming information. In addition, 56% of waiting rooms have visible indicators of adolescent or young adult support and friendliness to include images of adolescents or young adults on pictures and/or pamphlets.

**Table 4. Clinic workflow operations checklist.**

| Number of Clinics Reporting (%) | Clinical Indicator |
|---|---|
| 9 (100) | Providers conduct a sexual history. |
| | Patients are asked to provide urine for GC/CT NAAT. |
| 6 (67) | Patients self-collect specimens for GC/CT NAAT. |
| 8 (89) | Providers collect or request oropharyngeal and rectal specimens for GC/CT NAAT. |
| | Conduct patient satisfaction assessments after each visit, quarterly, and/or annually per CDS policy. |
| 5 (56) | Providers collect a genital specimen for GC/CT NAAT. |
| 7 (78) | Providers discuss HIV testing, if needed. |
| | Nurses or medical assistants conduct rapid point-of-care tests including pregnancy, HIV, syphilis, and GC/CT NAAT. |
| **Number of Clinics Reporting (%)** | **Non-Clinical Indicator** |
| 4 (44) | Have waiting rooms with visible indications of LGBTQ support such as a rainbow flag, a designated safe space sticker, images of same-sex couples on educational materials, and/or images of transgender affirming information. |
| 5 (56) | Have waiting rooms with visible indicators of adolescent or young adult support and friendliness to include images of adolescents or young adults on pictures and/or pamphlets. |

Codes from the *Clinical Team Member Interview* were inductively generated to fall into 1 of 3 intervention categories (training, clinical, and non-clinical) to guide theme generation. Resultant themes were triangulated with aggregate survey responses that further supported thematic evidence. The mixed method findings of the *Clinical Team Member Interview* are discussed below.

Six of the 9 CDSs conduct an annual STI screening, however 7 of the CDSs do not test more than once a year, unless the patient presents symptomatic for a bacterial STI. The interviews that were conducted emphasized the need for more bacterial STI training among all clinical team members. Even though annual screenings and testing are performed, it was identified that most of the clinical prescribers were not conducting a comprehensive sexual history. The lack of a consistent comprehensive sexual history has created missed opportunities for additional testing and may be contributing to the increased bacterial STI incidence rates in each CDS's respective jurisdiction.

Clinical team members stated that trainings were offered to their teams, however, it was noted that some of the training on cultural competency and stigma covered broad topics were not comprehensive nor taught new or useful knowledge. In addition, these training sessions were not offered to the entire clinic staff. It was evident that culturally competent non-clinical prescribers play an important role in a patient's experience and they should be included in these training sessions. One respondent stated, "We don't really have any training, unless we go (on our own)," while another stated, "We have done a little bit, but we know that's a weakness. So, in our monthly office meetings, we've gone on the internet and gotten the terms, just handed those out and talked about them a little bit. So, we've barely just scratched the surface. I'm hoping that we can do a lot more in that area, because it's a need. It very much is a need." Overall, all interviewees expressed a desire to participate in more cultural competency training.

The findings of the clinical component of the *Clinical Team Member Interview* showed that providers were not conducting consistent, comprehensive, sexual histories on intake across the CDSs. This was also demonstrated in the *Clinical Team Member Process*, *Attitudes & Beliefs*

*Survey*, where only 44% of respondents reported conducting a sexual history on intake. In addition, only 74% of providers were conducting follow-up sexual histories at acute care visits when a patient presented with symptoms of an STI. One respondent stated, "Some of my resident physicians feel awkward taking care of patients who are homosexual or transgender and they don't know how to ask these patients the right sexual history questions."

CDSs were not found to be routinely conducting GC/CT *NAATs* per CDC guidelines, or every 3–6 months, at 3 anatomical sites as applicable (genital/urine, oropharyngeal, and rectal). Sixty-seven percent (67%) of clinics tested for bacterial STIs on at least an annual basis; 18% tested for bacterial STIs every 3–4 months; and 78% tested for bacterial STIs if symptomatic for an STI. Fifty-nine percent (59%) of CDSs reported self-collection for oropharyngeal and rectal GC/CT NAAT specimens. One respondent stated, "They (patients) will complain about it but always allow us to do the swab. We always do the swab. Patients will complain of course saying this uncomfortable. I don't like this."

The findings of the clinical component of the *Clinical Team Member Interview* also showed that CDSs are currently task sharing among clinical and non-clinical personnel to meet clinic needs. However, task sharing is often unorganized. For example, among prescribing providers, 85% and 52% conduct sexual histories and collect NAAT specimens for GC/CT, respectively. Among non-prescribing providers, 59% and 44% conduct sexual histories and collect NAAT specimens for GC/CT, respectively. Task sharing was most evident among patient notification to return to the clinic following positive STI results as follows: nurses (74%), prescribing providers (59%), case managers (7%), patient navigators (4%), social workers (4%), and other providers (eg, DIS, MA) (30%). One respondent stated, "I think communication is needed between the team to make sure that they're on the lookout for those that need to just come back and retest." Finally, CDSs expressed funding needs to retain and hire additional clinical and non-clinical staff.

The non-clinical component found that CDSs could improve how welcoming their clinic space is and that provider comfort and stigma may be a barrier to care. One respondent stated, "There is stigma from the community and concern about perception of what others think; We have patients that don't want to be seen and will travel far distances to receive care at another clinic." On a scale of 1 to 5, (1) very unfriendly, (2) unfriendly, (3) neutral, (4) friendly, and (5) very friendly, results from the *Clinical Team Member Interview* demonstrated 25% of respondents rated their clinics as *less than friendly* (ie, scale of 1–3) to LGBTQ individuals, and 56% of clinic waiting rooms do not have visible indications of LGBTQ support.

## Discussion

The findings of this phase 1 mixed method needs assessment informed the selection of evidence-based interventions to be implemented in a phase 2 multi-site, multi-level evaluation study to improve STI screening and testing of bacterial STIs among people with or at risk for HIV. The results revealed a need for improved opportunities for provider cultural competency and bacterial STI screening and treatment training, increased routine (non-acute) bacterial STI screening and testing frequency, increasing the "welcoming" measures of each clinic to increase the engagement of those at highest risk of STIs, and making the process of routine screening (sexual history taking), testing, and follow-up as patient-centered as possible while limiting added burden on the clinical team members. Findings informed the selection of the following 4 interventions: (1) use of an audio computer-assisted self-interview (ACASI) sexual history; (2) patient self-collection of GC/CT NAAT specimens; (3) implementation of measures identified to make a clinic space more LGBTQ welcoming; and (4) provider training related to bacterial STI screening, testing, treatment, and follow-up.

Qualitative analysis showed that CDSs have trouble conducting consistent sexual histories and this gap may influence the frequency of STI tests being done. Indeed, 9 CDSs reported not conducting consistent (inter- and intra-clinic), comprehensive routine sexual histories and conducting routine testing (every 3–6 months) for those with identified risk for an STI as defined by the CDC. Quantitative analysis showed that 44% of the CDSs conduct a consistent, comprehensive sexual history on intake, and 18% test for STIs every 3–4 months. These findings support those of other studies that have found that comprehensive sexual health histories as part of routine care are not common [22]. Less than 40% of providers conduct sexual histories with patients, and many do not receive formal sexual history training in school [2]. In addition, previous studies have documented that only one third to one half of primary care clinicians routinely screen men or women for STIs [23].

Based on the need for consistent, comprehensive sexual health histories to determine which bacterial STI tests should be done, the use of an audio computer-assisted self-interview (ACASI) sexual history taken at each routine clinic visit (including lab visits) and appropriate acute care clinic visits was selected. Included at the end of the sexual history are questions on preference of self-collection or provider collected GC/CT NAAT specimens for each anatomical site if needed. At the end of the ACASI-based sexual history will be a summary of needed tests based on answers provided in the sexual history along with the patient's preference for self-collection or provider collected. It is hypothesized that increasing patient self-collection of GC/CT NAAT specimens may decrease patient refusal of testing as experienced by some providers. Since many HIV care providers request patients get their lab work done 1–2 weeks prior to a routine monitoring HIV clinic visit, the ACASI-based sexual health history will be administered before lab visits to see if any additional STI-related tests are indicated. A standing order will be set up in each of the CDSs to allow designated staff to order additional STI tests as needed when the patient completes the sexual history at a lab visit.

Increasing self-collection of GC/CT NAAT specimens by patients has several advantages including saving time for clinical providers and results that are non-inferior to provider-collected specimens. In addition, self-collection of NAAT specimens provides privacy and sensitivity that may be preferred by patients and allows the provider's time to be dedicated to addressing patient symptoms and needs [5, 24, 25]. ACASI allows for conducting consistent, routine sexual histories while improving the reliability of patient provided answers [20, 26, 27]. In addition, ACASI can inform providers' screening and treatment practices. Provider training related to bacterial STI screening, testing, treatment, and follow-up may also make it easier for providers to identify appropriate bacterial STI tests, treatment, and testing sites.

The LGBTQ welcoming clinic space intervention was selected because 26% of the respondents rated their clinics as less than friendly to LGBTQ individuals, and more than half of the clinic waiting rooms at the CDSs did not have visible indications of LGBTQ support. More welcoming clinic spaces may improve patient perception of cultural safety at the clinic and potentially improve new and existing patient-provider relationships [7, 8].

The *Clinical Team Member Process, Attitudes and Beliefs Survey* provided evidence of topics to be addressed through provider training. Those 4 topic areas will be addressed within the following trainings: (1) Epidemiology, Diagnosis, and Treatment; (2) Culturally Affirming Care to Reduce Stigma; (3) Taking a Comprehensive Sexual History; and (4) Success Stories on Improving STI Care. Provider training on cultural competence, as well as the implementation of a LGBTQ and adolescent/young adult welcoming clinic space are hypothesized to improve provider-patient engagement and increase comfort level of patients in being open with healthcare providers about sexual and gender identity. The literature cites lack of provider training on administering sexual health histories, as well as lack of provider comfort discussing sexual health histories [2]. Additionally, lack of culturally competent care has been cited as a large

barrier in receiving sexual health services, with recommendations for increased provider training in this area [15].

Limitations in selecting evidence-based interventions to be implemented at each of the 9 CDSs existed and included subjective intervention decision making without rigorous analytical methods for choosing the interventions. To not disrupt clinical operations, investigators limited online surveys and in-person interviews to 3 staff per clinic, a clinical prescriber, a clinical non-prescriber, and the designated Change Champions from each CDS for a total sample size of 27. While this pre-intervention mixed method needs assessment was designed to obtain comprehensive feedback from a multidisciplinary team at each CDS, in addition to a small sample size, clinical prescribers and clinical non-prescribers have heterogeneous backgrounds and roles that may have influenced individual responses. Another limitation was the difficulty in extracting historical data from the CDS electronic health records (EHRs). Five different EHRs were used by the 9 CDSs, and the extraction of baseline data was identified as challenging for each of the CDSs. For example, identifying how many patients had oropharyngeal GC/CT testing or rectal GC/CT testing done in the past year was challenging. Additionally, there was no uniform way to identify the number of transgender or nonbinary patients cared for in the clinic, nor percentage of gay/lesbian, heterosexual, bisexual/pansexual patients cared for. Creating data reports for the specific measures requested was difficult for each of the CDSs, regardless of EHR type used.

## Conclusion

In sum, this mixed method needs assessment to inform a large scale, multi-site, multi-level intervention and evaluation plan employed a minimally invasive approach to selecting evidence-based interventions to improve routine bacterial STI screening, testing, treatment and follow-up. The evaluation dimensions of each intervention will be analyzed individually via mixed methods at the aggregate, jurisdiction, and clinic level for the total study population and each subpopulation in the phase 2 study to evaluate interventions to enhance compliance with CDC recommendations for STI testing and treatment for populations disproportionately affected by STIs in and outside of HIV care.

## Supporting information

**S1 File. Clinical team member interview.**
(PDF)

**S2 File. Pre-intervention data survey (2016–2017).**
(PDF)

**S3 File. Clinical team member process, attitudes, & beliefs survey.**
(PDF)

**S4 File. STI screening readiness checklist.**
(PDF)

**S5 File. Clinic workflow operations checklist.**
(PDF)

**S6 File. Results of the clinical team member interview.**
(PDF)

**S7 File. Results of the pre-intervention data survey (2016–2017).**
(PDF)

**S8 File. Results of the clinical team member process, attitudes, & beliefs survey.**
(PDF)

**S9 File. Results of the STI screening readiness checklist.**
(PDF)

**S10 File. Results of the clinic workflow operations checklist.**
(PDF)

## Author Contributions

**Writing – original draft:** Kathleen Cullinen, Macsu Hill, Taylor Anderson.

**Writing – review & editing:** Veronica Jones, John Nelson, Mirna Halawani, Peijia Zha.

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
