## [Decision Letter · Decision Letter 0]

4 Sep 2021

PONE-D-21-13426

Improving sexually transmitted infection screening, testing, and treatment among people with HIV: A mixed methods needs assessment to inform a multi-site, multi-level intervention and evaluation plan

PLOS ONE

Dear Dr. Cullinen,

Thank you for submitting your manuscript to PLOS ONE. After careful consideration, we feel that it has merit but does not fully meet PLOS ONE’s publication criteria as it currently stands. Therefore, we invite you to submit a revised version of the manuscript that addresses the points raised during the review process.

We look forward to receiving your revised manuscript.

Kind regards,

Janet E Rosenbaum, Ph.D.

Academic Editor

PLOS ONE

Journal Requirements:

2. When reporting the results of qualitative research, we suggest consulting the COREQ guidelines: http://intqhc.oxfordjournals.org/content/19/6/349. In this case, please consider including more information on the number of interviewers, their training and characteristics; and please provide the interview guide used.

Reviewers' comments:

Reviewer's Responses to Questions

**Comments to the Author**

1. Is the manuscript technically sound, and do the data support the conclusions?

Reviewer #1: Yes

Reviewer #2: Yes

2. Has the statistical analysis been performed appropriately and rigorously? 

Reviewer #1: Yes

Reviewer #2: Yes

3. Have the authors made all data underlying the findings in their manuscript fully available?

Reviewer #1: Yes

Reviewer #2: Yes

4. Is the manuscript presented in an intelligible fashion and written in standard English?

Reviewer #1: Yes

Reviewer #2: Yes

5. Review Comments to the Author

Reviewer #1: 1. Line 197: The description of the table is confusing because it sounds like there should be a figure for both 1) the patient population and 2) the number of patients tested, diagnosed, and treated, unless it’s the case the entire patient population has received one or more services (i.e., testing, diagnosis, treatment).

2. Line 217: repeated words

3. Need a better description of the STI Screening Readiness Checklist, specifically what response categories were available (e.g., yes/no versus scaled). It’s difficult to interpret Table 3 without information related to response categories. Unsure what the “number of clinics reporting” means. Does it mean clinics said yes to the readiness item?

4. Should define the three inductive themes to introduce readers prior to going into detail about each.

5. Doesn’t seem necessary to separate the “mixed method section” from the prior descriptive statistics, given that the same statistics are repeated. Would suggest integrating the two sections to tell a cohesive story.

6. Discussion/conclusion: Do the results from participating clinics align with the literature? If so, in what ways? If not, how do they differ.

Reviewer #2: This manuscript addresses the critically important intersection of STI and HIV care and highlights difficulties in stemming the STI epidemic in the context of HIV care. The mixed methods approach is appreciated. Overall, the manuscript is well written and offers new insight, but there some aspects are less clear.

Abstract: Clear, well-written

Introduction: Background explained. The 3-year HRSA project is explained.

Methodology:

-What is meant by “key CDS staff?” How are the selected? What are the criteria for a Change Champion?

-What are the 9 evidence-based interventions? Please explain or provide reference.

-Importantly, the selected jurisdictions have very high STD and HIV rates. How many sites were in each jurisdiction?

-More clarity for the quantitative tools would be appreciated. Some of these instruments utilize scales but are other tools using Y/N response?

-There are only 3 respondents per site, which although may interfere less with clinic flow, it may not offer as robust representation. Additionally, non-prescribers (RN, SW) have very different backgrounds, so there is some heterogeneity introduced by this. Authors should address this and how participants are selected.

Results:

-Generally well written.

-Would be helpful to better explain the data for Table 1; in particular, are each of the persons in the table positive for an STI? The phrase “tested, diagnosed, and treated” is somewhat redundant if they’re all positive. Furthermore, more info regarding totals for each jurisdiction would be insightful.

-For quantitative assessment, it would helpful to know the breakdown of participant responses by category. This might be helpful to target interventions if there is a noticeable gap among prescribers, for instance.

-For Tables 3-5: were these quantitative tools administered to each of the 3 participants at each site, or were they administered to on site to the composite team?

Discussion/Conclusion:

-Well written and clear. Provides good discussion on the gaps and where interventions would fit. While the selected interventions seem to fit the gaps identified in the assessment, more information about the interventions available would be insightful.

6. PLOS authors have the option to publish the peer review history of their article (what does this mean?). If published, this will include your full peer review and any attached files.

Reviewer #1: No

Reviewer #2: No

---

## [Author Response · Author response to Decision Letter 0]

18 Oct 2021

October 18, 2021

Dear Dr. Rosenbaum:

Detailed below are the responses to each point raised by the academic editor and reviewers for the revision and resubmission of PONE-D-21-13426: Improving sexually transmitted infection screening, testing, and treatment among people with HIV: A mixed method needs assessment to inform a multi-site, multi-level intervention and evaluation plan.

ACADEMIC REVIEWER COMMENTS: 

Comment 1. Please ensure that your manuscript meets PLOS ONE's style requirements, including those for file naming. The PLOS ONE style templates can be found at https://journals.plos.org/plosone/s/file?id=wjVg/PLOSOne_formatting_sample_main_body.pdf and https://journals.plos.org/plosone/s/file?id=ba62/PLOSOne_formatting_sample_title_authors_affiliations.pdf

Response 1. The revised manuscript meets PLOS ONE's style requirements, including those for file naming. 

Comment 2. When reporting the results of qualitative research, we suggest consulting the COREQ guidelines: http://intqhc.oxfordjournals.org/content/19/6/349. In this case, please consider including more information on the number of interviewers, their training and characteristics; and please provide the interview guide used.

Response 2. More information on the qualitative research has been provided on Page 7 (ie, number of interviewers and their training and characteristics). The interview guide has been uploaded as a Supporting Information file (S1 File. Clinical Team Member Interview).

Comment 3. In your Data Availability statement, you have not specified where the minimal data set underlying the results described in your manuscript can be found. PLOS defines a study's minimal data set as the underlying data used to reach the conclusions drawn in the manuscript and any additional data required to replicate the reported study findings in their entirety. All PLOS journals require that the minimal data set be made fully available. For more information about our data policy, please see http://journals.plos.org/plosone/s/data-availability.

"Upon re-submitting your revised manuscript, please upload your study’s minimal underlying data set as either Supporting Information files or to a stable, public repository and include the relevant URLs, DOIs, or accession numbers within your revised cover letter. 

Response 3. This study’s minimal underlying data set has been uploaded as Supporting Information files (S6 File. Results of the Clinical Team Member Interview; S7 File. Results of the Pre-Intervention Data Survey (2016-2017); S8 File. Results of the Clinical Team Member Process, Attitudes, & Beliefs Survey; S9 File. Results of the STI Screening Readiness Checklist; S10 File. Results of the Clinic Workflow Operations Checklist).

REVIEWER COMMENTS: 

Reviewer #1: 1. Line 197: The description of the table is confusing because it sounds like there should be a figure for both 1) the patient population and 2) the number of patients tested, diagnosed, and treated, unless it’s the case the entire patient population has received one or more services (i.e., testing, diagnosis, treatment).

2. Line 217: repeated words

3. Need a better description of the STI Screening Readiness Checklist, specifically what response categories were available (e.g., yes/no versus scaled). It’s difficult to interpret Table 3 without information related to response categories. Unsure what the “number of clinics reporting” means. Does it mean clinics said yes to the readiness item?

4. Should define the three inductive themes to introduce readers prior to going into detail about each.

5. Doesn’t seem necessary to separate the “mixed method section” from the prior descriptive statistics, given that the same statistics are repeated. Would suggest integrating the two sections to tell a cohesive story.

6. Discussion/conclusion: Do the results from participating clinics align with the literature? If so, in what ways? If not, how do they differ.

Reviewer #2: This manuscript addresses the critically important intersection of STI and HIV care and highlights difficulties in stemming the STI epidemic in the context of HIV care. The mixed methods approach is appreciated. Overall, the manuscript is well written and offers new insight, but there some aspects are less clear.

Abstract: Clear, well-written

Introduction: Background explained. The 3-year HRSA project is explained.

Methodology:

-What is meant by “key CDS staff?” How are the selected? What are the criteria for a Change Champion?

-What are the 9 evidence-based interventions? Please explain or provide reference.

-Importantly, the selected jurisdictions have very high STD and HIV rates. How many sites were in each jurisdiction?

-More clarity for the quantitative tools would be appreciated. Some of these instruments utilize scales but are other tools using Y/N response?

-There are only 3 respondents per site, which although may interfere less with clinic flow, it may not offer as robust representation. Additionally, non-prescribers (RN, SW) have very different backgrounds, so there is some heterogeneity introduced by this. Authors should address this and how participants are selected.

Results:

-Generally well written.

-Would be helpful to better explain the data for Table 1; in particular, are each of the persons in the table positive for an STI? The phrase “tested, diagnosed, and treated” is somewhat redundant if they’re all positive. Furthermore, more info regarding totals for each jurisdiction would be insightful.

-For quantitative assessment, it would helpful to know the breakdown of participant responses by category. This might be helpful to target interventions if there is a noticeable gap among prescribers, for instance.

-For Tables 3-5: were these quantitative tools administered to each of the 3 participants at each site, or were they administered to on site to the composite team?

Discussion/Conclusion:

-Well written and clear. Provides good discussion on the gaps and where interventions would fit. While the selected interventions seem to fit the gaps identified in the assessment, more information about the interventions available would be insightful.

RESPONSES TO REVIEWER COMMENTS: 

REVIEWER #1: 

Comment 1. Line 197: The description of the table is confusing because it sounds like there should be a figure for both 1) the patient population and 2) the number of patients tested, diagnosed, and treated, unless it’s the case the entire patient population has received one or more services (i.e., testing, diagnosis, treatment).

Response 1. The description of the table has been revised on Page 10 to provide an estimate of the baseline numbers for the subpopulations tested, diagnosed, and/or treated for GC, CT and/or syphilis. The entire patient population did receive one or more services (i.e., testing, diagnosis, treatment) at each of the CDSs during 2016 and 2017.

Comment 2. Line 217: repeated words

Response 2. The repeated words “less than” were deleted on Page 11.

Comment 3. Need a better description of the STI Screening Readiness Checklist, specifically what response categories were available (e.g., yes/no versus scaled). It’s difficult to interpret Table 3 without information related to response categories. Unsure what the “number of clinics reporting” means. Does it mean clinics said yes to the readiness item?

Response 3. The response categories for the STI Screening Readiness Checklist were clarified on Page 8 as consisting of 1) Yes; 2) No; and 3) I don’t know for each given statement. Also, the label on Table 3 on Page 14 was edited to clarify that the responses represent the number of clinics reporting Yes for each indicator or given statement.

Comment 4. Should define the three inductive themes to introduce readers prior to going into detail about each.

Response 4. The definitions of the three inductive themes were to introduce readers prior to going into detail about each were provided on Pages 9-10.

Comment 5. Doesn’t seem necessary to separate the “mixed method section” from the prior descriptive statistics, given that the same statistics are repeated. Would suggest integrating the two sections to tell a cohesive story.

Response 5. The Mixed methods analysis section was relabeled as Results on Page 10. Also, the Mixed methods analysis subheading on Page 18 was deleted to integrate the two sections and to tell a cohesive story.

Comment 6. Discussion/conclusion: Do the results from participating clinics align with the literature? If so, in what ways? If not, how do they differ.

Response 6. Revisions/additions have been made to the describe how the results align with the literature on Pages 21-23.

REVIEWER #2: 

METHODOLOGY

Comment 1. What is meant by “key CDS staff?” How are the selected? What are the criteria for a Change Champion?

Response 1. The selection process for the key CDS staff and the criteria for the Change Champion are described on Page 5.

Comment 2. What are the 9 evidence-based interventions? Please explain or provide reference.

Response 2. The 9 evidence-based interventions are described on Pages 5-6 with references.

Comment 3. Importantly, the selected jurisdictions have very high STD and HIV rates. How many sites were in each jurisdiction?

Response 3. There were 3 sites selected within each of the 3 jurisdictions as described on Page 7.

Comment 4. More clarity for the quantitative tools would be appreciated. Some of these instruments utilize scales but are other tools using Y/N response?

Response 4. Clarity for the quantitative tools has been provided on Pages 8-9 including response categories/types. In addition, all assessment tools have been uploaded as Supporting Information files S1-S5.

Comment 5. There are only 3 respondents per site, which although may interfere less with clinic flow, it may not offer as robust representation. Additionally, non-prescribers (RN, SW) have very different backgrounds, so there is some heterogeneity introduced by this. Authors should address this and how participants are selected.

Response 5. Participant selection was detailed on Page 5. The limitation introduced by responses from clinical prescribers vs. clinical non-prescribers to obtain comprehensive feedback from a multidisciplinary team is addressed on Page 24. 

RESULTS

Comment 1. Would be helpful to better explain the data for Table 1; in particular, are each of the persons in the table positive for an STI? The phrase “tested, diagnosed, and treated” is somewhat redundant if they’re all positive. Furthermore, more info regarding totals for each jurisdiction would be insightful.

Response 1. The description of the table has been revised to provide an estimate of the baseline numbers for the subpopulations tested, diagnosed, and/or treated for GC, CT and/or syphilis overall and per each jurisdiction. The entire patient population did receive one or more services (i.e., testing, diagnosis, treatment) at each of the CDSs during 2016 and 2017.

Comment 2. For quantitative assessment, it would be helpful to know the breakdown of participant responses by category. This might be helpful to target interventions if there is a noticeable gap among prescribers, for instance.

Response 2. The breakdown of participant responses by category were provided with a cautionary note for interpretation as the Change Champion at each CDS may also serve as a clinical prescriber or clinical non-prescriber on Page 12.

Comment 3. For Tables 3-5: were these quantitative tools administered to each of the 3 participants at each site, or were they administered to on site to the composite team?

Response 3. On Pages 8-9, details were provided on the administration of the quantitative tools including mode of administration, location, and individuals vs. the composite team who completed them.

DISCUSSION/CONCLUSION

Comment 1. Well written and clear. Provides good discussion on the gaps and where interventions would fit. While the selected interventions seem to fit the gaps identified in the assessment, more information about the interventions available would be insightful.

Response 1. More information on the selected interventions and how these study findings relate to those in the literature are provided on Pages 21-23.

Thank you for the opportunity to revise and resubmit this manuscript.

Sincerely,

Kathleen Cullinen, Ph.D., MS, RDN

---

## [Decision Letter · Decision Letter 1]

13 Dec 2021

Improving sexually transmitted infection screening, testing, and treatment among people with HIV: A mixed method needs assessment to inform a multi-site, multi-level intervention and evaluation plan

PONE-D-21-13426R1

Dear Dr. Cullinen,

We’re pleased to inform you that your manuscript has been judged scientifically suitable for publication and will be formally accepted for publication once it meets all outstanding technical requirements.

Kind regards,

Janet E Rosenbaum, Ph.D.

Academic Editor

PLOS ONE

Additional Editor Comments (optional):

Reviewers' comments:

Reviewer's Responses to Questions

**Comments to the Author**

1. If the authors have adequately addressed your comments raised in a previous round of review and you feel that this manuscript is now acceptable for publication, you may indicate that here to bypass the “Comments to the Author” section, enter your conflict of interest statement in the “Confidential to Editor” section, and submit your "Accept" recommendation.

Reviewer #1: All comments have been addressed

Reviewer #2: All comments have been addressed

2. Is the manuscript technically sound, and do the data support the conclusions?

Reviewer #1: Yes

Reviewer #2: Yes

3. Has the statistical analysis been performed appropriately and rigorously? 

Reviewer #1: Yes

Reviewer #2: Yes

4. Have the authors made all data underlying the findings in their manuscript fully available?

Reviewer #1: Yes

Reviewer #2: Yes

5. Is the manuscript presented in an intelligible fashion and written in standard English?

Reviewer #1: Yes

Reviewer #2: Yes

6. Review Comments to the Author

Reviewer #1: (No Response)

Reviewer #2: (No Response)

7. PLOS authors have the option to publish the peer review history of their article (what does this mean?). If published, this will include your full peer review and any attached files.

Reviewer #1: No

Reviewer #2: No

---

## [Editor Report · Acceptance letter]

16 Dec 2021

PONE-D-21-13426R1 

Improving sexually transmitted infection screening, testing, and treatment among people with HIV: A mixed method needs assessment to inform a multi-site, multi-level intervention and evaluation plan 

Dear Dr. Cullinen:

I'm pleased to inform you that your manuscript has been deemed suitable for publication in PLOS ONE. Congratulations! Your manuscript is now with our production department. 

Kind regards, 

on behalf of

Dr. Janet E Rosenbaum 

Academic Editor

PLOS ONE